# Price trends of reimbursed oncological drugs in Switzerland in 2005–2019: A descriptive analysis

Yael Rachamin[1], Christoph Jakob Ackermann[2], Oliver Senn[1], Thomas Grischott[1]*

1 Institute of Primary Care, University of Zurich and University Hospital Zurich, Zurich, Switzerland,
2 Department of Medical Oncology, Spital STS AG, Thun, Switzerland

☯ These authors contributed equally to this work.
* thomas.grischott@usz.ch

**Data Availability Statement:** All relevant data are within the manuscript and its Supporting Information files.

**Funding:** The authors received no specific funding for this work.

## Abstract

Increasing oncological treatment costs are a major global concern with the risk of entailing two-tiered health care. Among cost determining factors is the price of individual drugs. In recognition of the central role of this factor, we present a comprehensive overview of the development of monthly prices of oncological drugs introduced over the last 15 years in Switzerland. We identified all oncological drugs newly reimbursed by mandatory health insurance in 2005–2019, and searched public repositories for their package prices, indications with approval dates, and treatment regimens for the calculation of (indication-specific) monthly prices. We found 81 products covering 77 different substances (39.5% protein kinase inhibitors, 21.0% monoclonal antibodies). Most indications related to the topography "blood", followed by "lung and thorax" and "digestive tract". From 2005–2009 to 2015–2019, the median monthly product price over all distinct indications of all products decreased by 7.56% (CHF 5,699 [interquartile range 4,483–7,321] to CHF 5,268 [4,19–6,967]), whereas it increased by 73.7% for monoclonal antibodies. In December 2019, six products had monthly prices over CHF 10,000, all approved for hematological or dermatological cancers. Our analysis suggests that individual price developments of oncological drugs are presently not the major driver of rising cancer treatment costs. However, rising launch prices of some new, mostly hematological drugs are of concern and require continued monitoring.

## 1. Introduction

The advent of new oncological drugs, particularly checkpoint inhibitors and many other targeted compounds, revolutionized cancer care in the past few years and led to an unprecedented improvement of prognosis for many patients worldwide. However, increasing oncological drug costs are a major global concern with the risk of entailing two-tiered health care [1, 2]. In 2010 already, cancer drugs were estimated to make up for 10% of drug costs in high-income countries [1], trending to further increase [3].

**Competing interests:** The authors have declared that no competing interests exist.

Rising health care expenditure for cancer drugs can be explained by increasing costs per patient or by a growing demand. Demand is driven by increasing cancer incidence and prevalence due to demographic developments, more abundant risk factors, progress in early disease detection, and declining cancer mortality rates due to more efficient treatments [3–5]. Costs per patient in turn depend on drug prices as well as other factors such as treatment duration, increasingly common drug combination treatment regimens or switches from cheaper to more expensive products [6, 7].

It is not a priori clear how this multitude of intertwining factors impacts the overall expenditure for cancer drug treatment, neither on the health system level nor per patient. However, prices of the individual oncological drugs are a key factor that deserves a particular focus, firstly because of its public visibility which, accentuated by increasingly opaque pricing, frequently leads to media headlines [8], and secondly and more importantly so because, contrary to other determinants, drug prices are subject to external control through political and regulatory authorities [3, 9, 10]. In this context, it is worth mentioning that regulators are increasingly being called upon to take cost-benefit considerations into account in price negotiations [1, 11], and in order to answer this call, detailed knowledge of the prices and price trends of individual drugs is a necessary prerequisite.

Most recent studies on oncology drug pricing reported increasing launch [6, 12–14] and post launch prices [6, 10, 15–17]. Interestingly, one very recent analysis of cancer drug prices in the United States and Europe found that launch prices increased in all countries considered, whereas post launch prices increased in the United States but decreased in several European countries including Switzerland [18]. However, none of these studies considered extended indications, and a detailed analysis of cancer drugs price trends in Switzerland, with stratification for important subgroups of tumors and substances, is currently missing.

With this study, we present a comprehensive overview of the development of monthly prices of oncological drugs subjected to mandatory reimbursement by health insurers in the last 15 years in Switzerland, and contrast it with recent developments in comparator countries and with previous price trends in Switzerland itself, in order to contribute a piece of the puzzle of the rising total health care expenditure on cancer drugs. It is not our intention to re-prove the already all-too-well-known fact that total oncological treatment costs are rising [6, 19], but instead we aim at investigating whether price developments of individual drugs–both trends in initial approval prices as well as subsequent price developments after approval–are a dispositive driver of the overall cost trends.

## 2. Methods

We analyzed monthly prices of cancer drugs in Switzerland for which mandatory reimbursement by basic health insurance began within the fifteen-year period from January 2005 to December 2019. Since some of these drugs are used for more than one condition (each of which might in turn be treated following different regimens) and, on the other hand, some drugs are produced by multiple manufacturers (each of whom might sell the same substance under several different brand names and usually in a variety of galenic forms and/or package sizes), some clarification of terminology is necessary in order to allow accurate description of methodology (i.e. the calculation of monthly indication-specific product prices).

### 2.1. Definitions and selection of cancer drugs

We searched publicly available monthly snapshots of the Federal Office of Public Health's so-called "Specialities List" (SL) of all drugs under reimbursement by mandatory basic health insurance in Switzerland [20]. Each SL entry (henceforth called *"preparation"*) is characterized

by brand name, galenic formulation, unit and amount of active ingredient, and package size. For each preparation, the SL contains the public and ex-factory prices and additional information such as its Anatomical Chemical Classification (ATC) code [21] with corresponding generic substance name (INN), and the "Index Therapeuticus" (IT; a drug classification system used by the Swiss Agency for Therapeutic Products, Swissmedic [22]).

We identified preparations relating to cancer by their IT because ATC codes were only introduced into the SL in December 2015. We included all preparations with an IT of the form 07.16.zz ("Oncologica") that were first included into the SL within the last 15 years (January 2005–December 2019), and excluded preparations with ATC codes outside the therapeutic subgroups L01 ("antineoplastic agents") and L02 ("endocrine therapy") as well as preparations of the chemical subgroup L01XD ("sensitizers used in photodynamic/radiation therapy"). Preparations with identical brand names were grouped into what will henceforth be called *"products"* (disregarding different doses and galenic formulations), while the metagroups *"substances"* were distinguished by their International Nonproprietary Names (INNs).

## 2.2. Identification and classification of indications

For each included product, we identified all approved indications with corresponding approval dates using the Swiss drug compendium [23] or, if information was missing, the Swissmedic journals [24]. All identified indications, of which we disregarded cancer stage and treatment line, are listed in the online S1 Table, column "Indication". The indications were further categorized by *"topography"* (S1 Table, column "Topography") essentially following the International Statistical Classification of Diseases and Related Health Problems, 10th Revision (ICD-10), Chapter 2 ("Neoplasms") [25]. Indications for benign tumors were ignored.

## 2.3. Calculation of monthly product prices

Based on the public prices specified in the SL, raw *indication-specific* monthly product prices (henceforth referred to as *"monthly product prices"*) were calculated as the minimum unit price over all preparations of a specific product, multiplied by the indication-specific monthly dose. Monthly doses were calculated based on the current (if not available: latest) treatment regimens according to the Swiss drug compendium [23] for an adult patient of 70 kg with a body surface of 1.7 m$^2$, and assuming months of 30 days. From several possible therapeutic regimens for the same indication we chose the one with the fewest additional components and, secondarily, the regimen with the lowest monthly dose. We disregarded costs for premedication and all concomitant therapies (e.g. pain medication), and also ignored dose adjustments for comorbidities and optional adjustments depending on treatment success. In case of time-dependent treatment regimens (e.g. with ramp-ups or induction cycles) we calculated the mean monthly doses for the plateau phase (S1 Table, column "Monthly dose"). In a sensitivity analysis to assess the impact of inflation–at 4.6% overall low in Switzerland from 2005 to 2019 –we also adjusted monthly product prices to their December 2019 values using monthly consumer price index (CPI) data from the Swiss Federal Statistical Office [26].

## 2.4. Data analysis

We used medians with interquartile ranges (IQR) as well as numbers (*n*) and proportions (%) to report the results. Time trends of monthly product prices were expressed as percentage increases between the median monthly product prices over the first five years (2005–2009) and the median monthly product prices over the last five years (2015–2019). In addition to overall time trends, the same calculation was repeated for each topography (S1 Table) and chemical subgroup as defined by ATC code level 4. Due to few observations in most subgroups, we

discriminated only monoclonal antibodies and protein kinase inhibitors and subsumed all other substances under "others" (S1 Table, column "Chemical subgroup"). All statistical analyses and plots were produced with the R statistical software, version 3.5.1 [27].

## 3. Results

We identified 81 different products first included in the SL in 2005–2019, covering 77 different substances. The products' substances belonged to the chemical subgroups of protein kinase inhibitors in 39.5% ($n = 32$), monoclonal antibodies in 21.0% ($n = 17$) and to "others" in the remaining 39.5% ($n = 32$). The products had a median of 1 indication (IQR = 1–2) over their observed reimbursement period (median 6 years, IQR = 3–9). The numbers of distinct indications of all individual products summed up to 141 overall (S1 Table). The most common topography was "blood" (33.3%, or 47 of all 141 indications), followed by "lung and thorax" and "digestive tract" (Table 1).

For the monthly product prices we found a median (over all distinct indications of all products over all months) of CHF 5,337 and an IQR of CHF 4,039–6,928. The median monthly product prices decreased by 7.56% from 2005–2009 (CHF 5,699, IQR = 4,483–7,321) to 2015–2019 (CHF 5,268, IQR = 4,019–6,967). Fig 1 shows all indication-specific monthly product prices (i.e., of all products with all indications) graphically over the entire observation period, and the corresponding numbers can be found in the appendix (S1 Table). Also in the appendix, we present an inflation-adjusted version of Fig 1 to demonstrate the negligible effect of inflation (S1 Fig).

### 3.1. Most expensive products

Within the whole observation period, 5 products (with a total of 6 indications) had maximum monthly product prices of over CHF 20,000: blinatumomab, inotuzumab ozogamicin, ipilimumab, nelarabine, and talimogene laherparepvec, all but ipilimumab (skin) treating hematological tumors. Three of these products (blinatumomab, inotuzumab ozogamicin and talimogene laherparepvec) were introduced in the last five years (2015–2019). A maximum monthly product price exceeding CHF 10,000 was observed for 11 products (17 indications): blinatumomab,

**Table 1. Median monthly product prices and price trends by topography.**

| Topography | ICD-10 block | Number of indications (of reimbursed products) | Median monthly product prices (IQR) | Change in median monthly product prices[a] |
|---|---|---|---|---|
| Lip, oral cavity, pharynx | C00–C14 | 1 (1) | 8,501 (7,412–9,066) | NA[b] |
| Digestive tract | C15–C26 | 15 (11) | 5,104 (4,083–5,703) | -12.9% |
| Lung and thorax | C30–C39 | 19 (16) | 4,592 (3,400–5,650) | +7.6% |
| Skin | C43–C44 | 14 (14) | 8,019 (6,206–11,571) | NA |
| Connective and soft tissue | C45–C49 | 4 (3) | 10,018 (4,039–10,018) | NA |
| Breast | C50 | 11 (11) | 4,171 (3,476–6,530) | -12.1% |
| Female genital organs | C51–C58 | 6 (3) | 5,620 (3,930–6,530) | NA |
| Male genital organs | C60–C63 | 5 (5) | 4,298 (227–5,130) | NA |
| Urinary tract | C64–C68 | 12 (11) | 5,392 (4,508–6,530) | -27.2% |
| Brain | C69–C72 | 1 (1) | 7,859 (6,530–8,167) | -23.5% |
| Endocrine glands | C73–C75 | 6 (5) | 4,978 (4,425–6,162) | NA |
| Blood | C81–C96 | 47 (28) | 5,396 (3,709–7,362) | -17.7% |

ICD-10: International Statistical Classification of Diseases and Related Health Problems,10[th] Revision; IQR: interquartile range. [a] from the first five years (2005–2009) to the last five years under consideration (2015–2019). [b] NA: no price trend was calculated because there were no reimbursable products included in the SL in 2005–2009.

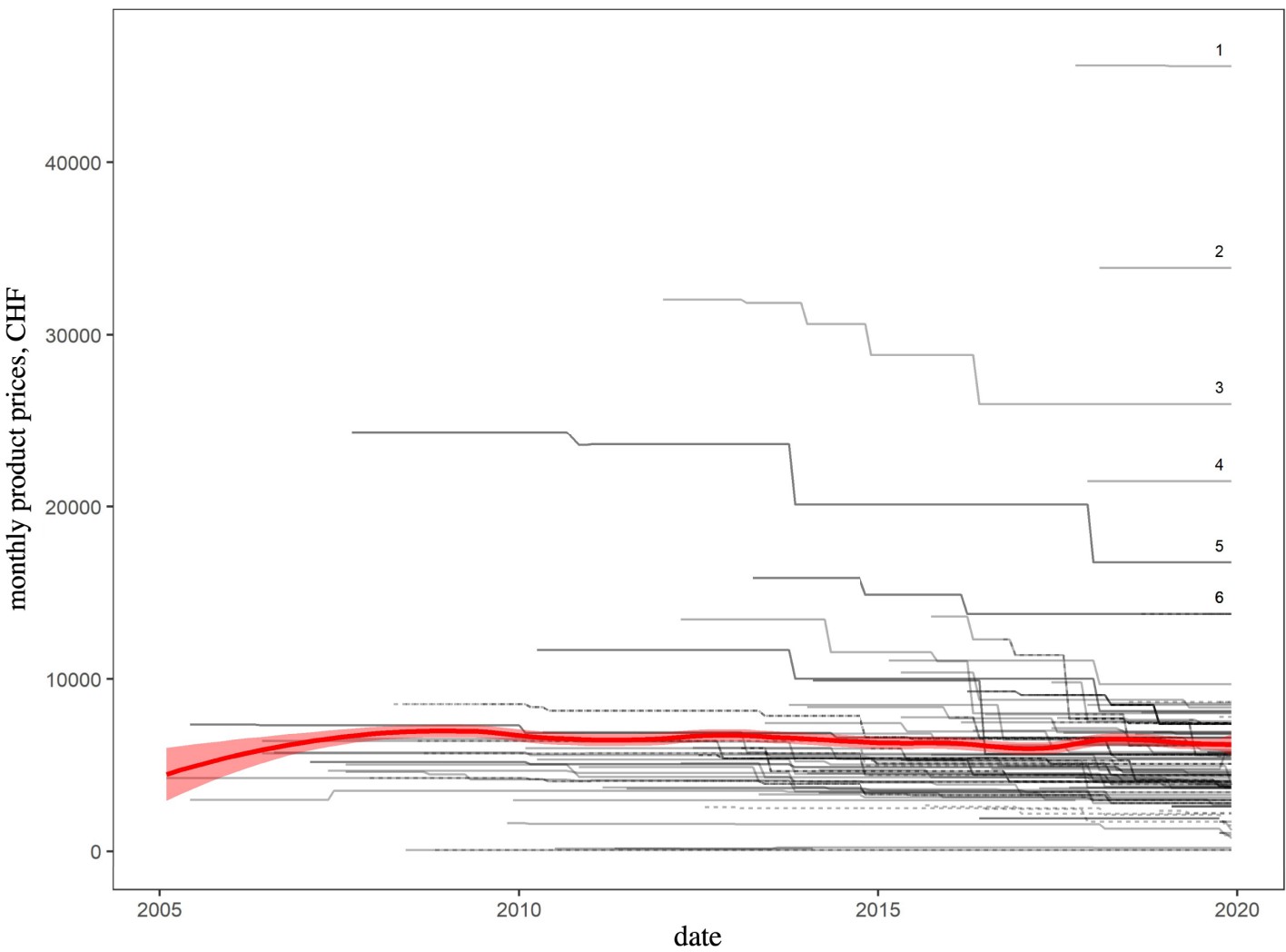

**Fig 1. Monthly product prices over time.** Each of the 141 black lines represents one specific indication of one specific product. Solid lines = first indications, dashed lines = subsequent indications. Products with monthly prices > CHF 10,000 in December 2019 are labelled with Arabic numbers: [1] blinatumomab (indicated for Acute lymphoblastic leukemia), [2] inotuzumab ozogamicin (Precursor B-cell acute lymphoblastic leukemia), [3] ipilimumab (Melanoma), [4] talimogene laherparepvec (Melanoma), [5] nelarabine (T-cell acute lymphoblastic leukemia or T-cell lymphoblastic lymphoma), [6] brentuximab vedotin (Morbus Hodgkin, Systemic anaplastic large-cell lymphoma, Cutaneous T-cell lymphoma, or Peripheral T-cell lymphoma). Local regression line with uncertainty band in red (loess [28] with span = 0.25 and pointwise ± 1.96 standard error).

inotuzumab ozogamicin, ipilimumab, nelarabine, talimogene laherparepvec, brentuximab vedotin, pembrolizumab, vemurafenib, trabectedin, arsenic trioxide, and ibrutinib. Six of these products were introduced in the last five years (blinatumomab, inotuzumab ozogamicin, talimogene laherparepvec, pembrolizumab, arsenic trioxide, and ibrutinib). At the end of the observation period (December 2019), 6 products (with a total of 10 indications; Fig 1) had monthly product prices over CHF 10,000 (blinatumomab, inotuzumab ozogamicin, ipilimumab, talimogene laherparepvec, nelarabine, brentuximab vedotin).

## 3.2. Monthly product prices by chemical subgroup

Comparing monthly product prices among chemical subgroups showed that monoclonal antibodies were more expensive than protein kinase inhibitors, with median monthly product

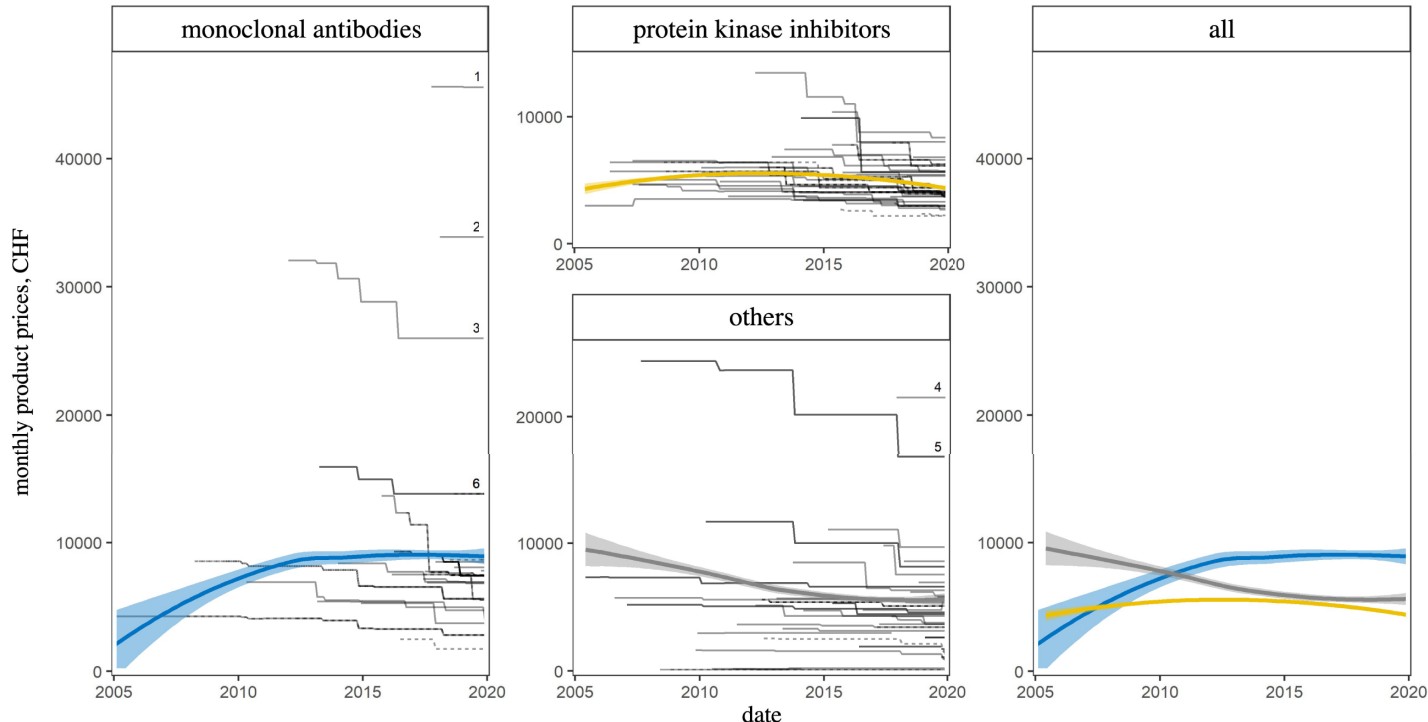

**Fig 2. Monthly product prices over time by chemical subgroup.** Each of the 141 black lines represents one specific indication of one specific product. Solid lines = first indications, dashed lines = subsequent indications. Local regression lines with uncertainty bands in color (loess [28] with span = 1 and pointwise ± 1.96 standard error). Numeric product labels as in Fig 1.

prices of CHF 7,412 (IQR = 5,281–8,501) versus CHF 4,674 (IQR = 4,058–5,965). The median monthly product prices of monoclonal antibodies increased by 73.7% between 2005–2009 and 2015–2019 whereas they decreased by 22.1% for protein kinase inhibitors (Fig 2).

### 3.3. Monthly product prices by topography

Table 1 shows median monthly product prices together with price trends from the first five years (2005–2009) to the last five years under consideration (2015–2019), grouped by topography (medians with IQRs calculated over all distinct indications of all products within each topography and over all relevant months). Of all six topographies with related products included into the SL in the first five years (2005–2009), all but one ("lung and thorax") saw a decrease in monthly product prices (Table 1 and Fig 3). For the remaining topographies (lip, oral cavity, pharynx; skin; connective and soft tissue; female and male genital organs; endocrine glands), no new products were included in the first five years (2005–2009).

### 4. Discussion

Based on publicly available pricing data of oncological drugs newly reimbursed through basic health insurance in Switzerland in the last 15 years, we observed that, while launch prices of part of the drugs started from increasingly higher levels, subsequent price adjustments after SL inclusion generally resulted in lower prices. As a consequence, monthly drug prices remained relatively stable overall (Fig 1), and price trends for different topographies tended to converge towards a range of approximately CHF 4,000–10,000 (Fig 3, last facet).

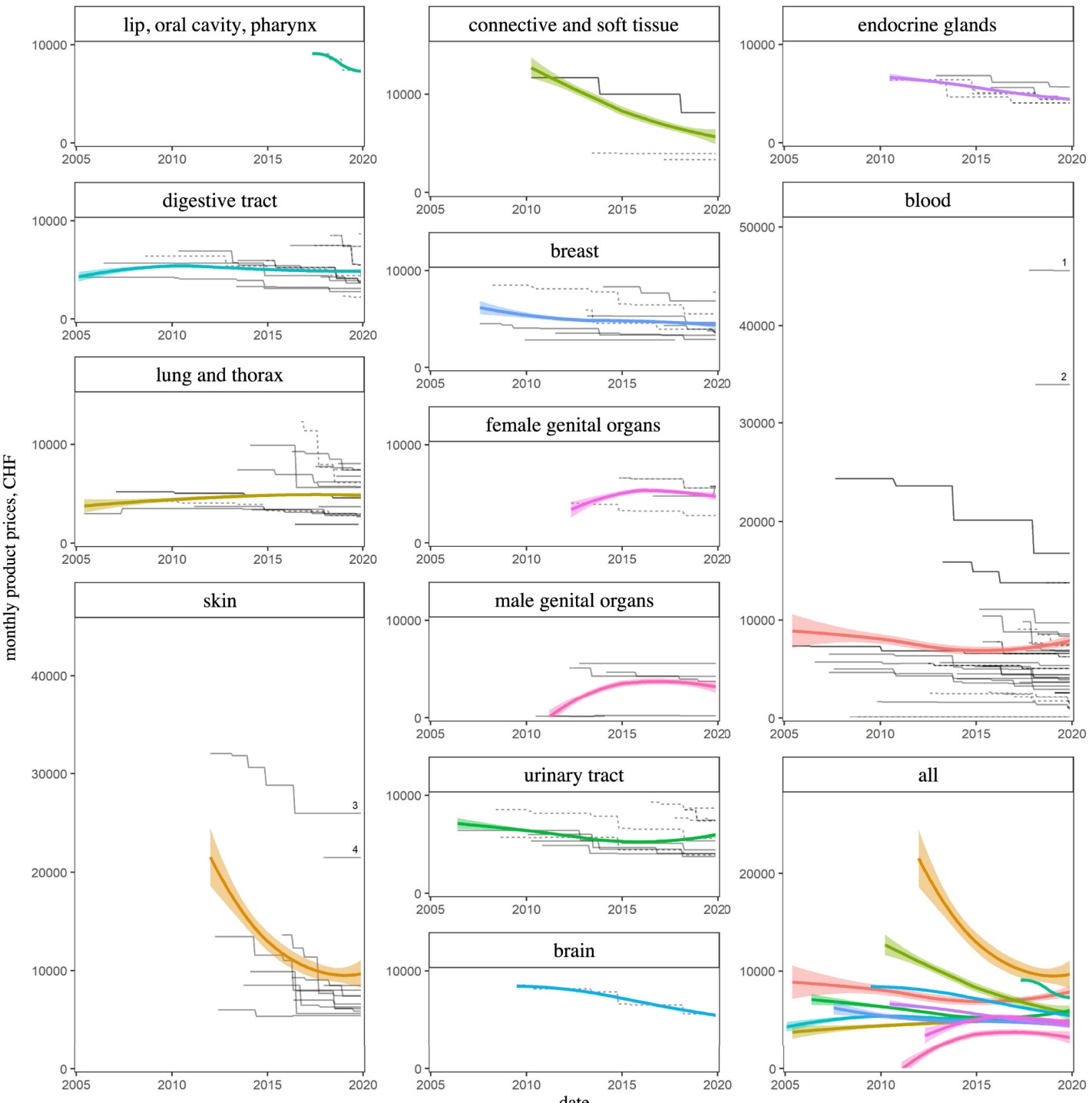

**Fig 3. Monthly product prices over time by topography.** Each of the 141 black lines represents one specific indication of one specific product. Solid lines = first indications, dashed lines = subsequent indications. Local regression lines with uncertainty bands and numeric product labels as in Fig 2.

## 4.1. Initial vs. post launch price trends

Average drug prices at any given time are the result of developments of both initial and post launch prices. Most of the recent investigations of drug price developments stem from the

United States and report increasing launch [6, 12–14] and post launch prices [6, 10, 15–17] of individual oncological drugs. Obviously, these findings cannot simply be transferred to other countries, as the context of the prevailing regulation policy must be considered. In the United States, where, due to strong advocacy for an unregulated free market system, cancer drug prices are essentially freely set by the manufacturers without any regulatory boundaries. In Switzerland, as in other high-income European countries including Germany, the United Kingdom and France, prices of drugs that are to be reimbursed under mandatory public health insurance are negotiated between the government or a designated entity and the manufacturer or pharmaceutical industry [3, 10]. This applies to both initial prices which are set before (e.g. in Switzerland) or shortly after (e.g. in Germany) [9] the new drugs are accepted for reimbursement, and post launch prices which are reviewed and–if deemed necessary–adjusted periodically (e.g. every 5 years in the United Kingdom [10], or every 3 years in Switzerland [29]). Savage et al. showed that initial cancer drug prices were approximately 42% higher at launch in the United States than in the United Kingdom, and that whereas there was an annual post launch price rise of about 8.8% in the United States, post launch prices have not increased in the United Kingdom over the last few years [10]. Vokinger et al. found an even larger discrepancy between the United States and selected European countries when comparing monthly treatment costs for recent oncological drugs [30]. In a recent study, the group also showed that while launch prices increased in all countries, post launch prices increased in the United States but decreased in several European countries including Switzerland [18]. In the present study, we additionally show that as a combined effect of rising launch and decreasing post launch prices, average monthly drug prices in Switzerland remained relatively stable overall.

The observed decrease in post launch prices may surprise, given the fact that Switzerland has seen steep post launch price increases of several cancer drugs in the past, e.g. following their sale to new owners [31]. Among potential explanations are currency effects: The Swiss franc gained about 39% compared to the Euro from 2005 to 2019 [32], and exchange rates influence drug prices in Switzerland via external reference pricing, which is an element of the aforementioned periodic review of pharmaceutical prices. However, the fact that similar downward trends in post launch cancer drug prices were recently observed in countries of the Euro area as well [18] makes it unlikely that the trend in Switzerland should be attributed to currency effects alone, but a detailed discussion of possible reasons is outside the scope of this article.

## 4.2. Indication extensions

Extensions of indications have rarely been considered in research on cancer drug prices. In our study, they allowed for finer classifications and thus more accurate calculation of median price trends within chemical subgroups and topographies.

Multiple indications can influence drug prices in several ways. First, a recent literature review [33] identified several models of indication-specific pricing (ISP), i.e. different prices for multiple indications of the same substance produced by the same manufacturer. In our analysis, we found identical unit prices for all indications per substance and manufacturer. Obviously, this results in monthly drug treatment costs which depend on specific dosage schemes (and on the availability of generic drugs from other manufacturers) but not on indication-specific clinical benefits. However, since we did not consider rebates, we cannot assess whether variable benefits are reflected in indication-specific rebates.

Second, extensions of indications may have an impact on the drug price for existing indications, e.g., as a result of expanding sales volume or if a weighted mean price is chosen across all

indications, with weights reflecting benefits in each separate indication. For 7 products we detected price reductions within two months after indication extensions (afatinib: indication extended 15 November 2017 to squamous cell carcinoma of the lung; atezolizumab: 3 May 2019 urothelial carcinoma; bevacizumab: 17 November 2014 cancer of the fallopian tube and primary peritoneal carcinoma; nivolumab: 19 April 2018 colorectal cancer; pembrolizumab: 22 March 2016 melanoma, 11 October 2016 non-small-cell lung cancer; ruxolitinib: 25 September 2015 polycythemia vera; sorafenib: 12 November 2014 thyroid cancer). It is impossible to say though to what extent these price reductions were actually related to the newly added indications, or whether they were due to other reasons considered in the periodic price review.

### 4.3. Volume effects and most expensive outliers

National and OECD data show that both the costs of oncological treatments for individual patients as well as the total expenditure on cancer treatment have increased over the period considered [34, 35]. Our results indicate that the isolated price developments of the majority of oncological drugs recently accepted for mandatory reimbursement cannot be the main reason. Instead, increasingly common treatment combinations of expensive drugs and treatment switches from cheaper to more expensive products, including such not on the SL, are most likely among the key cost drivers. By revealing that price trends of most newer oncological drugs did not fuel the alarming increase in cancer treatment costs, our analysis suggests that such volume effects might affect overall costs even more than expected.

It was estimated that in 2011, cancers of the digestive tract (including colorectal and gastric cancers) were the most expensive cancers in Switzerland (with total direct costs of 1501 million CHF), followed by lung cancer (721 million CHF) and cancers of the male genital organs (including prostate cancer) (433 million CHF) [36]. The topographies with the lowest estimated costs were urinary tract (241 million CHF), brain (224 million CHF), and skin (221 million CHF). Table 1 shows the largest relative price reductions in topologies with total costs at the lower end of the range, but there is no clear association of price trends with total treatment costs over all topologies, nor with the number of newly reimbursed products per topology.

Of particular interest in the context of the overall rising cancer treatment costs are the most expensive oncological drugs currently on the SL, namely blinatumomab, inotuzumab ozogamicin, ipilimumab, talimogene laherparepvec, nelarabine, brentuximab vedotin (Fig 1). The majority of these drugs are antibodies (Fig 2) for hematological indications (Fig 3). Even though their indications are still relatively narrow, these expensive drugs constitute reasons for concern, as our study shows that (unit) prices usually did not drop to markedly lower levels with or shortly after indication extensions (S1 Table) and it must therefore be feared that these expensive drugs' indications may expand at unchanged prices. Moreover, the availability of similarly priced drugs will most probably increase in the future following scientific progress. It is therefore worthwhile to investigate whether the prices of these substances are–and will be–fully justified by a particularly high clinical benefit.

### 4.4. Strengths and limitations

Our analysis is the first detailed long-term exploration of time trends of cancer drug prices in Switzerland. To our knowledge, it is the first study of cancer drug prices, even beyond Switzerland, that assesses price developments after initial pricing taking subsequent indications into account. We consider as particular strength of our work that it presents unwieldy archive data in an illustrative way and thus makes it accessible for interpretation in the first place. It is not meant as a concluding work but should rather shed light on a highly relevant and publicly discussed topic, where regulatory control is possible but comprehensible data are scarce.

Limitations of our study are: We analyzed list prices of reimbursed drugs only, and ignored both rebates as well as older drugs still available but included into the SL more than 15 years ago (i.e. before 2005). We approximated monthly doses and disregarded treatment duration and combination therapies. Also, for simplicity, our classification of indications was rather generic, ignoring therapy lines and disease stages (which are known to affect prices [37]).

## 5. Conclusion

Overall, monthly prices of most newly reimbursed drugs in 2005–2019 moved within a price range of approximately CHF 4,000–10,000 with a median of CHF 5,300. However, some new, mostly hematological drugs escape this pattern. Continuing research is needed to investigate how these outliers–and also future drugs–will impact future cancer treatment costs, but presently it seems that the price development of individual products is not the major driver of rising cancer treatment costs as increasing initial prices are met by decreasing post launch prices.

## Supporting information

**S1 Table. Yearly prices of oncological drugs newly reimbursed by mandatory basic health insurance in 2005–2019 in Switzerland.**
(XLSX)

**S1 Fig. Monthly product prices over time, inflation-adjusted to their December 2019 values.** Each of the 141 black lines represents one specific indication of one specific product. Solid lines = first indications, dashed lines = subsequent indications. Local regression line with uncertainty band and numeric product labels as in Fig 1.
(EPS)

## Author Contributions

**Conceptualization:** Yael Rachamin, Christoph Jakob Ackermann, Thomas Grischott.

**Data curation:** Yael Rachamin, Thomas Grischott.

**Formal analysis:** Yael Rachamin, Thomas Grischott.

**Methodology:** Yael Rachamin, Thomas Grischott.

**Project administration:** Thomas Grischott.

**Supervision:** Oliver Senn, Thomas Grischott.

**Validation:** Yael Rachamin, Thomas Grischott.

**Visualization:** Yael Rachamin.

**Writing – original draft:** Yael Rachamin, Thomas Grischott.

**Writing – review & editing:** Yael Rachamin, Christoph Jakob Ackermann, Oliver Senn, Thomas Grischott.

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
