## [Decision Letter · Decision Letter 0]

6 Oct 2021

PONE-D-21-30520Price trends of reimbursed oncological drugs in Switzerland in 2005–2019: a descriptive analysisPLOS ONE

Dear Dr. Thomas Grischott,

Thank you for submitting your manuscript to PLOS ONE. After careful consideration, we feel that it has merit but does not fully meet PLOS ONE’s publication criteria as it currently stands. Therefore, we invite you to submit a revised version of the manuscript that addresses the points raised during the review process.

We look forward to receiving your revised manuscript.

Kind regards,

Wen-Wei Sung, M.D., Ph.D.

Academic Editor

PLOS ONE

Reviewers' comments:

Reviewer's Responses to Questions

**Comments to the Author**

1. Is the manuscript technically sound, and do the data support the conclusions?

Reviewer #1: Yes

Reviewer #2: Yes

2. Has the statistical analysis been performed appropriately and rigorously? 

Reviewer #1: Yes

Reviewer #2: Yes

3. Have the authors made all data underlying the findings in their manuscript fully available?

Reviewer #1: Yes

Reviewer #2: Yes

4. Is the manuscript presented in an intelligible fashion and written in standard English?

Reviewer #1: Yes

Reviewer #2: Yes

5. Review Comments to the Author

Reviewer #1: This manuscript is well organized and some issue should be revised before publication.

1. Consider shortening the conclusion in the abstract.

2. In the introduction, the fourth paragraph is too long. Would you revise it and focus on main goal?

3. The figures should be edited to be more clear or, somewhat, attractive. Current ones look like the ones print screen from software without further editing.

Reviewer #2: This is a well-written manuscript that demonstrates a long-term exploration of the time trends in the price of anticancer drugs in Switzerland and attempts to find out what are the key cost drivers of the overall cost trend.

I believe that this study will help policies respond to the increasing cancer drug costs, and seek to identify whether these drugs are justified by a particularly high clinical benefit, which ultimately determines how best to systematically improve the underuse of high-value drugs and minimize the use of low-value drugs.

I have no issues to highlight in terms of content and statistical analysis. However, the figures and graphics in this manuscript are a bit messy for me. I think the author needs to think about how to make your artwork more clutter-free and visually appealing next time.

In short, I think this article is suitable for publication in PLOS ONE.

6. PLOS authors have the option to publish the peer review history of their article (what does this mean?). If published, this will include your full peer review and any attached files.

Reviewer #1: No

Reviewer #2: No

---

## [Author Response · Author response to Decision Letter 0]

27 Oct 2021

Reviewer #1: This manuscript is well organized, and some issue should be revised before publication.

- We would like to thank reviewer #1 for his positive opinion and suggestions on how to improve our article. We hope that our changes will meet your approval.

1. Consider shortening the conclusion in the abstract.

- We removed what could be considered as part of the results section and focused on interpretation and implication of the results. Page 2, lines 52-54.

2. In the introduction, the fourth paragraph is too long. Would you revise it and focus on main goal?

- Since the analysis is about drug price developments in Switzerland, we downweighed the comparison with the US and shortened the paragraph accordingly. The shortened paragraph now stresses the contrast of decreasing post-launch prices as observed in our current - and another recent – study with the overall situation of mostly rising launch and post-launch prices. We did not fully refrain from giving a very short overview over the literature since this was a request of a previous reviewer. Page 3, lines 84-87.

3. The figures should be edited to be more clear or, somewhat, attractive. Current ones look like the ones print screen from software without further editing.

- We added pointwise uncertainty bands (± 1.96 standard errors) to all figures for improved clarity and visually improved all figures by removing the plain gray boxes and by using the more appealing Times font. We hope that this will mitigate the impression of print screens, and we will gladly implement additional “beautification” according to both reviewers’ specific suggestions!

Reviewer #2: This is a well-written manuscript that demonstrates a long-term exploration of the time trends in the price of anticancer drugs in Switzerland and attempts to find out what are the key cost drivers of the overall cost trend.

I believe that this study will help policies respond to the increasing cancer drug costs, and seek to identify whether these drugs are justified by a particularly high clinical benefit, which ultimately determines how best to systematically improve the underuse of high-value drugs and minimize the use of low-value drugs.

I have no issues to highlight in terms of content and statistical analysis. However, the figures and graphics in this manuscript are a bit messy for me. I think the author needs to think about how to make your artwork more clutter-free and visually appealing next time.

In short, I think this article is suitable for publication in PLOS ONE.

- The authors would also like to thank reviewer #2 for appreciating our manuscript, in particular content and analysis. May we refer the reviewer to our above answer regarding figures and graphics? We would like to emphasize once again that we will gladly revise the figures further, according to specific requests.

---

## [Decision Letter · Decision Letter 1]

2 Nov 2021

Price trends of reimbursed oncological drugs in Switzerland in 2005–2019: a descriptive analysis

PONE-D-21-30520R1

Dear Dr. Thomas Grischott,

We’re pleased to inform you that your manuscript has been judged scientifically suitable for publication and will be formally accepted for publication once it meets all outstanding technical requirements.

Kind regards,

Wen-Wei Sung, M.D., Ph.D.

Academic Editor

PLOS ONE

Reviewers' comments:

Reviewer's Responses to Questions

**Comments to the Author**

1. If the authors have adequately addressed your comments raised in a previous round of review and you feel that this manuscript is now acceptable for publication, you may indicate that here to bypass the “Comments to the Author” section, enter your conflict of interest statement in the “Confidential to Editor” section, and submit your "Accept" recommendation.

Reviewer #1: All comments have been addressed

Reviewer #2: All comments have been addressed

2. Is the manuscript technically sound, and do the data support the conclusions?

Reviewer #1: Yes

Reviewer #2: (No Response)

3. Has the statistical analysis been performed appropriately and rigorously? 

Reviewer #1: Yes

Reviewer #2: (No Response)

4. Have the authors made all data underlying the findings in their manuscript fully available?

Reviewer #1: Yes

Reviewer #2: (No Response)

5. Is the manuscript presented in an intelligible fashion and written in standard English?

Reviewer #1: Yes

Reviewer #2: (No Response)

6. Review Comments to the Author

Reviewer #1: (No Response)

Reviewer #2: (No Response)

7. PLOS authors have the option to publish the peer review history of their article (what does this mean?). If published, this will include your full peer review and any attached files.

Reviewer #1: No

Reviewer #2: No

---

## [Editor Report · Acceptance letter]

4 Nov 2021

PONE-D-21-30520R1 

Price trends of reimbursed oncological drugs in Switzerland in 2005–2019: a descriptive analysis 

Dear Dr. Grischott:

I'm pleased to inform you that your manuscript has been deemed suitable for publication in PLOS ONE. Congratulations! Your manuscript is now with our production department. 

Kind regards, 

on behalf of

Dr. Wen-Wei Sung 

Academic Editor

PLOS ONE